# Whole body physiology model to simulate respiratory depression of fentanyl and associated naloxone reversal
Austin Baird ®[1] ✉, Steven A. White[2], Rishi Das[2], Nathan Tatum[2] & Erika K. Bisgaard[3]

## Abstract

**Background** Opioid use in the United States and abroad is an endemic part of society with yearly increases in overdose rates and deaths. In response, the use of the safe and effective reversal agent, naloxone, is being fielded and used by emergency medical technicians at a greater rate. There is evidence that repeated dosing of a naloxone nasal spray is becoming more common. Despite this we lack repeated dosing guidelines as a function of the amount of opiate the patient has taken.

**Methods** To measure repeat dosing guidelines, we construct a whole-body model of the pharmacokinetics and dynamics of an opiate, fentanyl on respiratory depression. We then construct a model of nasal deposition and administration of naloxone to investigate repeat dosing requirements for large overdose scenarios. We run a single patient through multiple goal directed resuscitation protocols and measure total naloxone administered.

**Results** Here we show that naloxone is highly effective at reversing the respiratory symptoms of the patient and recommend dosing requirements as a function of the fentanyl amount administered. We show that for increasing doses of fentanyl, naloxone requirements also increase. The rescue dose displays a nonlinear response to the initial opioid dose. This nonlinear response is largely logistic with three distinct phases: onset, rapid acceleration, and a plateau period for doses above 1.2 mg.

**Conclusions** This paper investigates the total naloxone dose needed to properly reverse respiratory depression associated with fentanyl overdose. We show that the current guidelines for a rescue dose may be much lower than required.

## Plain Language Summary

Opioids such as fentanyl are a type of drug that reduce pain. However, the overdose of opioids causes severe breathing issues that can lead to death. Overdose of opioids is an increasing problem across the globe, particularly among people with opioid use disorder. To prevent death, first responders can administer a drug called naloxone that rapidly reverses the effects of opioids. However, the optimum amount of naloxone to administer is unclear. We use a mathematical model to investigate the effect of administering different amounts of naloxone during fentanyl overdose. Our findings suggest that the amount of naloxone to administer that is currently usually administered may be insufficient. Further research should enable naloxone usage guidelines to be optimized, which could improve survival following opioid overdose.

The opioid epidemic continues to kill countless people across the globe. In the United states alone 47,600 people have died in 2018[1], this reflects an increase of 5,100 since 2015[2]. Due to the amount of people dying to opioid-related overdose there has been a concerted effort to equip all emergency medical technicians to carry nasal naloxone as an effective reversal agent[3]. This equipment and subsequent increase in naloxone administration has been directly shown to decrease opioid-related deaths in controlled settings and specific regions[4–6]. Despite the efforts to properly equip first responders with the appropriate reversal agents, opioid-related deaths are still shockingly high and are showing a marked increase in countries around the world[7]. Throughout this progression of deaths there has been an associated increase of naloxone administration, with a 75% increase seen in between

the years 2016–2021[8]. This increase in administration mirrors the increase in opioid-associated overdose deaths during that period. The coupling of naloxone administration to opioid-associated overdose deaths is hypothesized to be related to more synthetic opiate use[9], more opiate use in the general population[10], inadequacies in naloxone reversal dosing[11], or compounding physiologic factors that are due to the type of opioid being used[12].

Synthetic opiates have become prominent in America over the past decade, particularly illicitly manufactured fentanyl[9]. During this, time increase in synthetic opiate deaths have outpaced traditional prescription opioids[10]. The nature of the pandemic has evolved as the type and potency of the drug has changed. This has placed a burden on first responders to administer more naloxone. There is a consensus that naloxone reversal

[1]University of Washington Department of Surgery, Division of Healthcare Simulation Sciences, Seattle, WA, USA. [2]Applied Research Associated Southeast Division, Raleigh, NC, USA. [3]University of Washington Department of Surgery, Division of Trauma, Burn, and Critical Care Surgery, Seattle, WA, USA. ✉e-mail: abaird1@uw.edu

dosing is inadequate to properly rescue many patients[11]. Studies have shown that overdose due to fentanyl may require more naloxone than what is currently in a single dose of Narcan (the most common form of reversal agent equipped to EMTs around the country)[13]. Despite these studies, many counties recommend administering the traditional amount of naloxone, not considering the evolving nature of the opioid epidemic[14]. Indeed, in response to this suffering, much of the United States opioid education and harm reduction strategy includes distributing naloxone directly to citizens. Although promising, there is evidence that this response isn't sufficient due to the higher naloxone dosing requirements for most illicitly manufactured fentanyl derivatives[15]. Due to the evolving nature of the issue, more understanding regarding dosing requirements for different amounts of administered fentanyl during illicit use is needed. This is particularly true to the for common EMT nasal spray form of naloxone, Narcan 2 and 4 mg doses.

To quantify and analyze the dose requirements and needs of first responders in relation to nasal naloxone administration, we develop a whole-body physiological model that can properly capture different dosing amounts, physiological respiratory depression, and model the interaction between a fentanyl and naloxone in the body. To date, various models have constructed nasal deposition and absorption models, with much of the work focused on 3D models of deposition[16,17]. Other work has been done to understand pharmacokinetic profiles of a drug in the blood after nasal administration[18–21]. Generally, these models consider breaking the nasal cavity down into a series of compartments with transfer rates corresponding to the drug being considered or the transport mechanism being modeled. Despite this, there are no models that consider dynamic interactions between the patient physiology, nervous system response, and repeated naloxone dose and interaction with fentanyl.

The software architecture of BioGears[22] allows for refined questions to be asked regarding naloxone administered as a nasal spray and its relation to the physiology of the patient experiencing fentanyl overdose symptoms. Here we describe a general pharmacokinetic model of fentanyl, using the BioGears engine, and a kinetic model of naloxone administered via a nasal spray device. We also describe a model of competitive mu-receptor drug interaction between fentanyl and naloxone. We consider fentanyl respiratory depression via a pharmacodynamic model that is coupled to a whole-body nervous system that captures large-scale regulatory responses in the body. Using these models, we investigate naloxone dosing requirements as a function of the initial fentanyl dose, given as a bolus. We consider the respiratory depression associated with fentanyl overdose and find that the rescue dose of naloxone to be much higher than is currently being deployed with first responders.

## Methods
The BioGears Engine fluid transport model constructs the analogous circuit model to describe the fluid dynamics of the cardiopulmonary system. We characterize the fluid and gas flows, pressures, and volumes by their electrical equivalents; current, voltage and capacitance, respectively. Circuit analogues have been used frequently in physiology modeling, usually to model cardiac pumping dynamics using a windkessel model[23–25]. BioGears expands this construction by using a series of connected three-element windkessels and other more sophisticated circuit architectures to model blood flow in each organ and in major vessels[22]. The software architecture of BioGears constructs objects representing circuit elements, allowing for organs, like the brain and kidney, to have larger, more sophisticated circuit structures Supplementary Fig. 1.

Pumping of the heart occurs by adjusting the compliance of the cardiac windkessel, effectively simulating contraction of the heart chambers. The BioGears respiratory system contains models defining gas exchange and diffusion. Collectively, these circuits constitute a lumped parameter, or zero-dimensional, system. That is, because no spatial component exists, the pressure, volumes, and flows calculated on each circuit represent values averaged (lumped) by each organ, vessel, and tissue component. Such an approach is appropriate for a model of this size considering the computational cost incurred by increasing fidelity[26]. Higher dimensional models require numerical solutions of either the full Navier-Stokes equations, or a simplification assuming, for instance, radial symmetry or low Reynolds number[27,28]. We choose this vasculature and cardiopulmonary fluid transport model to simulate and investigate naloxone reversal over hours postopioid overdose. Higher fidelity models would not be sufficient for the time scales considered for this research.

The BioGears physiology Engine organizes the data generated by the circuit structure and the various other connected models, representing the state of the patient, hierarchically into compartments. Top-level compartments generally represent organs or systems, with sub-compartments representing the vascular, tissue, extracellular, and intracellular spaces. The compartment structure is computationally organized as a graph to increase data processing speed. This graph is then leveraged to simulate the movement of substances in the body. Once the circuit is calculated and flow rates computed, the compartment graph then moves substances as a function of these flow rates and checks for mass conservation, Supplemental Note 1. Some transport connections require a more refined model, such as diffusion from the alveolar gas compartment into the alveolar capillaries. These connections are computed separately after the general substance movement is resolved.

The engine implements a Compartment Manager class that organizes and keeps track of compartment graphs and hierarchal mappings. Each simulation cycle, the state of the circuit is solved using a modified nodal analysis algorithm[29]. BioGears leverages the third-party numerical software Eigen 3 to compute the matrix inversion using a sparse LU factorization[30,31]. Substance flux, diffusion, and mass transfer is computed from the flow rates and tissue properties using the compartment graph and manager class. Various system and subsystem models are computed using a basic forward Euler scheme, for non-stiff equations. Custom implementation of an improved Euler method using two functional evaluations per time step is used to compute differential equations with more rapid dynamics. Each system implements its own class, providing the user the ability to directly implement methods specific to a certain organ. The system classes implemented in BioGears contain all the code used and described for this paper, specifically the drugs, respiratory, and nervous class files. We omit other model descriptions here for

brevity but refer the reader to the open-source documentation a for a full description.

The circuit and graph constructs are the backbone of the physiology engine with numerous physiological models (such as a robust nervous system) developed on top of, and often influencing, this backbone[32,33]. These models include a nervous system with baroreceptor and chemoreceptor feedback that modifies cardiovascular and respiratory activity; an active transport model that maintains ionic gradients across intracellular and extracellular compartments; a physiologically-based pharmacokinetic/pharmacodynamic (PBPK/PD) model that tracks the concentration-effect profile of numerous drugs[34]; a gastrointestinal model that determines rates of nutrient digestion and oral drug absorption; a renal feedback model that regulates urine production and substance filtration and reabsorption; and a metabolic consumption and production method that determines the energy demands of each organ. In addition, numerous injury models have been developed to influence the state of the patient ranging from acute hemorrhage to diabetes. Taken all together this platform provides the unique opportunity to not only investigate the kinetics of the drug in the body but also understand how fentanyl overdose impacts the physiology and oxygen transport in the body. We note that fentanyl administration effects multiple systems in the body with the state of the patient determined by feedback occurring in models connected the nervous, respiratory, and circulatory systems, Fig. 1. We will leverage this existing cardiopulmonary and nervous implementation to investigate naloxone reversal timing and dosing requirements for different fentanyl overdose situations.

### Nasal Administration Model
Pharmacokinetic and Transport Model: Circulation of substances in BioGears is driven by a lumped model of the cardiopulmonary system,

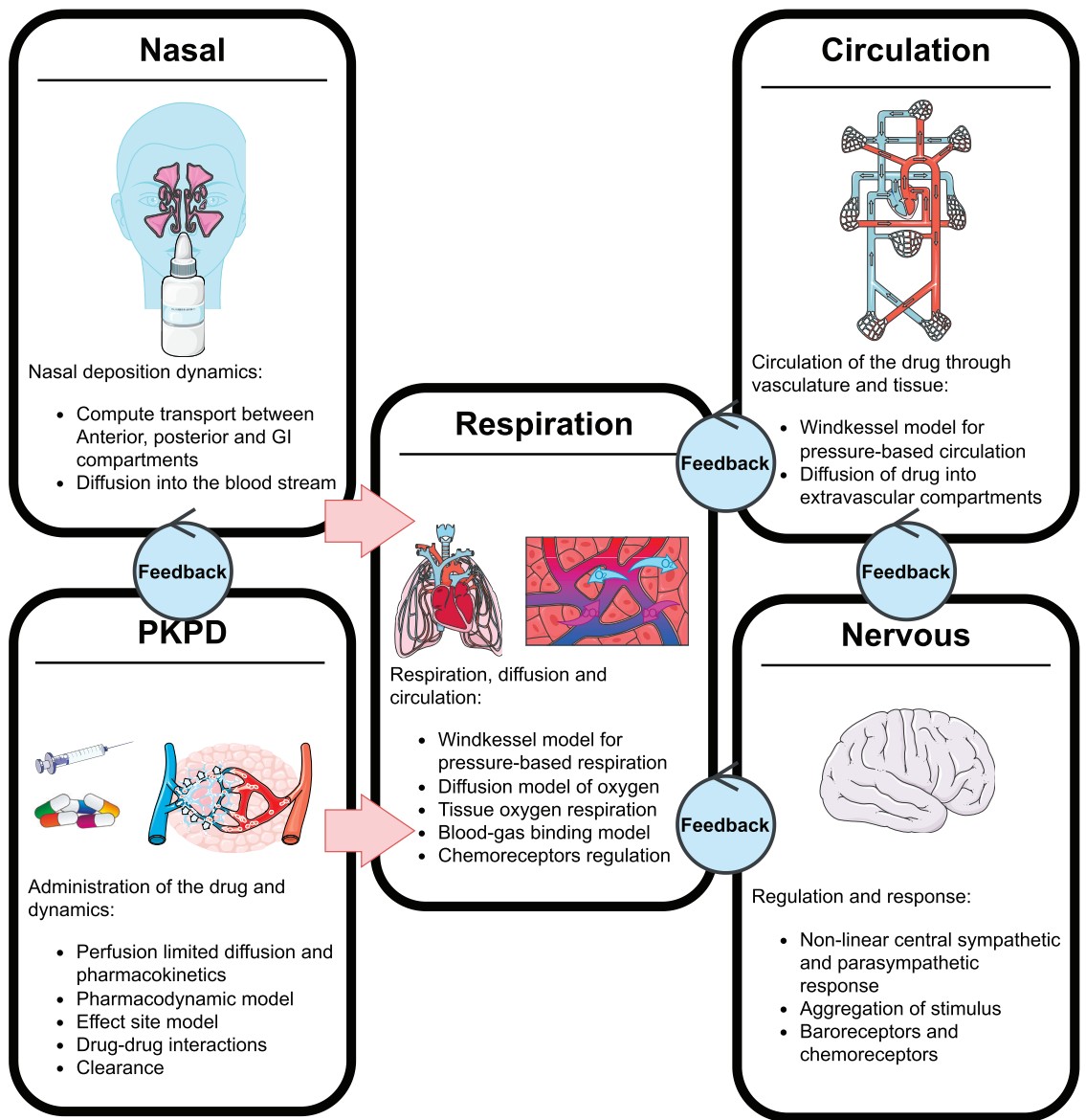

**Fig. 1 | The models in the physiology engine that respond to Fentanyl administration (venous bolus dose).** Feedback indicates that the two models interact with one another each time step, such as the drug-drug interaction model between Fentanyl and Naloxone. Figure created using BioIcons open-source icons. Icons used in the nasal figure are copyrighted under license CC 3.0. PKPD pill icon is licensed under CC 0, the syringe and vessels are licensed under CC 3.0, Respiration icons are provided under the CC 3.0 license, Nervous and circulation are similarly provided under CC 3.0 license.

which is based on prior physiological modeling. Drug administration models the substance mass in the appropriate compartment, depending on the administration route: intravenous, intraarterial, nasal, oral, intramuscular, and inhaled dose. Oral, nasal, and inhaled dose each consider deposition, digestion, and absorption into the appropriate tissue. For example, oral doses are absorbed into the vascular system via the small intestine compartment and inhaled doses are deposited into the lung tissue and absorbed into circulation via the alveolar compartment. For nasal naloxone we consider the compartmental structure of the nasal cavity with appropriate absorption metrics. Upon entering the circulation, we use perfusion-limited diffusion to deposit the drug in the various tissue compartments as it circulates. We expand upon prior work to compute the appropriate partition coefficient for a given tissue and substance[35,36]. Hence, for a given drug in circulation, we compute the change in mass over time as the concentration gradient for a given blood and tissue

compartment, scaled by a computed partition coefficient:

$$\frac{dm}{dt} = Q_B \left( C_B - \frac{C_T}{K_{TB}} \right) \tag{1}$$

Given a substance's physiochemical properties, Supplemental Table 1, we can compute the partition coefficient, as:

$$K_{TB} = \frac{f_u}{\gamma_{BP}} \left( f_{EW} + \frac{1 + 10^{pKa-pH_{IW}}}{1 + 10^{pKa-pH_P}} f_{IW} + \frac{k_a[AP^-]_T 10^{pKa-pH_{IW}}}{1 + 10^{pKa-pH_P}} \right.$$
$$\left. + \frac{P \cdot f_{NL} + (0.3P - 0.7)f_{NP}}{Y} \right) \tag{2}$$

For strong bases. Here we note that $f_u$ scales the classical formulation from plasma to blood. Similarly, for acids, neutral compounds, and weak

bases, we have

$$K_{TB} = \frac{f_u}{\gamma_{BP}}\left(f_{EW} + \frac{X}{Y}f_{IW} + \frac{P \cdot f_{NL} + (0.3P - 0.7)f_{NP}}{Y}\right.$$
$$\left. + K_{Bind}\left(\frac{1}{f_u} - 1 - \left(\frac{P \cdot f_{NL} + (0.3P - 0.7)f_{NP}}{Y}\right)\right)\right)$$ (3)

And for zwitter ions we have

$$K_{TB} = \frac{f_u}{\gamma_{BP}}\left(f_{EW} + \frac{X}{Y}f_{IW} + \frac{P \cdot f_{NL} + (0.3P - 0.7)f_{NP}}{Y}\right.$$
$$\left. + \left(\frac{Ka_{AP}[AP^-]_{EW}10^{Ka_{BASE}-PH_{IW}} - 10^{PH_{IW}-pKa_{ACID}}}{Y}\right)\right)$$ (4)

Here the term $[AP^-]_{EW}$ is the concentration of acidic phospholipids in the extracellular water and $Ka_{AP}$ is the association constant for a drug for acidic phospholipids. These three terms allow for computation of a given partition coefficient for a specific tissue and drug combination. The parameters X and Y depend on the state of the drug as follows:

$$X = \begin{cases} 1 + 10^{pH_{IW}-pK_a} \text{ Acid} \\ 1 + 10^{pK_a-pH_{IW}} \text{ Weak Base} \\ 1 \text{ Neutral} \end{cases}$$
$$Y = \begin{cases} 1 + 10^{pH_p-pK_a} \text{ Acid} \\ 1 + 10^{pK_a-pH_p} \text{ Weak Base} \\ 1 \text{ Neutral} \end{cases}$$ (5)

Parameters $pH_P$ and $pH_{IW}$ represent the pH of the plasma and intracellular water, respectively. Each parameter is computed at each time step in the simulation and will respond to changes in pH during a simulation if certain injuries present themselves to the patient. For this application they are effectively constant.

Clearance of the drug from the bloodstream occurs by hepatic and renal elimination. Here we define hepatic clearance by

$$Cl_H = \frac{Q_H f_u Cl_I}{Q_H + f_u Cl_I}$$ (6)

Here $Q_H$ represents the volumetric rate of blood supplied to the liver by the portal vein. In BioGears the small intestine, large intestine and splanchnic compartments send blood flow into the portal vein. $Cl_I$ represents intrinsic clearance and is normalized by the patient body weight. We then compute the mass of drug metabolized and removed from the liver compartment at each time step:

$$m_H(t) = BW \cdot Cl_H \cdot C_L(t) \cdot \Delta t$$ (7)

We define $Cl_L(t)$ as the concentration of the drug in the liver vasculature at time t. Other clearance elimination mechanisms are supported by Biogears, including renal, fecal, and systemic and are computed similarly.

We compute the pharmacodynamics of an opiate circulating in the body by computing an effect site concentration of the drug:

$$\frac{dC_e}{dt} = k_e(C_p - C_e)$$ (8)

This effectively models the transport of the drug from the systemic circulation to the site of action, or binding site. Here ke is a first-order rate constant, buffering the binding of the drug to the effect site. Cp and Ce are the concentration of the drug in the plasma and the effect site. To compute the pharmacodynamics of the drug, or the physiological response in the body, we define the variable E as the effect of the drug. We model E, the effect of the drug, by creating a functional relationship between an opiate, for

this study fentanyl, with its antagonist, naloxone:

$$\Delta E = E_b \frac{E_{max}C_e^\eta}{EC_{50}^\eta(1 + \frac{C_i}{k_i})^\eta + C_e^\eta}$$ (9)

Here Eb is the baseline of the given physiological effect the drug is exerting on the patient, such as cardiovascular or pulmonary effects. Emax is the maximal possible effect, EC50 is the half-maximal effect, and lastly Ci and ki are the inhibitor concentration and constant. For example, in the case of fentanyl, the inhibition concentration is the concentration of naloxone in the effect compartment. There are numerous possible effects that can be mapped to a specific drug in the BioGears physiology engine, Supplementary Table 1. This generic method of drug modeling has shown good correlation for a number of different drugs, given that the physiochemical properties detailed in Supplementary Table 2 are known and accurately measured in the literature[37]. The way the drug is administered will affect the concentration profile in the blood and our model will consider diffusion and release of the drug through the various nasal compartments to enter the bloodstream.

Nasal deposition model: Nasal drug administration offers an effective alternative drug delivery route by utilizing the large surface area of the nasal canals and the porous endothelial membrane, which lines them. The unique physiology of the nasal canal offers a vasculature space, which clears drugs rapidly[38]. Much like the intravenous or intramuscular delivery methods, nasal drug delivery skips the first-pass effect of the metabolism[39]. However, it provides similar benefits to oral dosing in that the nasal canal is a non-invasive pathway for rapid systemic drug absorption. While the narrow and nonlinear structure of the nasal canal can cause some challenges in drug retention, the nasal delivery method provides an excellent opportunity for drug exposure in a readily accessible system. In addition, most emergency rescue equipment kits include at least one form of nasal rescue spray, we will focus this paper on nasal delivery methods. We note, that the BioGears physiology engine also supports intramuscular, intravenous, and intraartial administrations in addition to the nasal route described here.

We define our nasal drug delivery model by building upon existing pharmacokinetic and transport models[16,19,20]. Following drug release, we track penetration and permeation through the nasal mucosa (modeled by anterior-posterior nasal compartments). By tracking the amount of drug released through each layer. We construct a series of first-order differential equations to solve for the drug quantities excreted through the gastro-intestinal tract and the amount absorbed into the systemic circulation. In BioGears, drug deposited into systemic circulation is then be managed by the substance transporter and existing pharmacokinetics and dynamics models.

We relate the transition of a drug administered through the nasal cavity to its transport through the anterior (A), posterior (P), and gastrointestinal (G) compartments:

$$\frac{-dA}{dt} = \alpha A$$ (10)

$$\frac{-dP}{dt} = -k_1 A + \beta P$$ (11)

$$\frac{-dG}{dt} = -k_5 P + \gamma G$$ (12)

$$\frac{-da}{dt} = -k_2 A + \delta a$$ (13)

$$\frac{-dp}{dt} = -k_6 P - k_3 a + \omega p$$ (14)

$$\frac{-dg}{dt} = -k_9 G - k_7 p + \in g$$ (15)

Where:

$$\alpha = k_1 + k_2 \tag{16}$$

$$\beta = k_5 + k_6 \tag{17}$$

$$\gamma = k_9 + k_{14} \tag{18}$$

$$\delta = k_3 + k_4 + k_{11} \tag{19}$$

$$\omega = k_7 + k_8 + k_{12} \tag{20}$$

$$\in = k_{10} + k_{13} + k_{15} \tag{21}$$

To produce the rate of drug input into systemic circulation:

$$R = k_4 a + k_8 p + k_{10} g \tag{22}$$

We note that the difference between upper case and lower casing for this model delineates between drug unreleased in a carrier solution and drug released from that carrier. We further note that different concentrations of nasal solutions could be modeled by modifying the fraction released in the anterior compartment. For the simulations detailed here, we model Narcan with 2 mg of naloxone suspended in a 0.1 mL solution. In BioGears, we update our vector of released and unreleased nasal drug quantities in each compartment (A, P, G, a, p, g). We solve this system of equations and multiply by the timestep to get the amount of drug mass to be moved into our vasculature compartments. We only consider the mass of the drug at each compartment in the nasal canal. We do not consider the changes in deposition due to the concentration of the administered spray, as there is some evidence that the concentration of the spray does have an effect on the kinetics of the administered dose[40]. We justify the use of this three-compartment model by noting that nasal deposition dynamics are generally quantified with a slower uptake, and delayed clearance, prompting the need for a more complex mathematical model[20,21,41,42].

Supplementary Table 3 details the First order rate constant (1/s) used in the ODE's for modeling nasal drug administration. Rate constants taken from[20,43]. Minor changes have been made in order to fit the model to the specific naloxone pharmacokinetic profiles of experimental data, whereas this model was first produced as a generic model of nasal administration.

## Central and peripheral nervous system model

Perturbations to oxygenation in the BioGears system are regulated and respond through a robust nervous system model. Efferent and Afferent responses are managed and weighted in the central nervous (CNS) system model. The afferent baroreceptor, chemoreceptor, and pulmonary stretch receptors are computed and collected by the central nervous system, which weights these responses and generates the efferent response on the appropriate system in BioGears. The chemoreceptor control model is based upon prior model development that considered hypoxic conditions on the nervous system[44–47], we connect these models to our model of the respiratory driver function, pharmacodynamic, and transport models in order to elicit a patient response depending on a variety of perturbations to the patient physiology. The chemoreceptors are modeled as two signals: a central and peripheral response, and their contributions are independent and additive, for the frequency adjustments to ventilation, we have:

$$f_t = f_{base} + \Delta f_c + \Delta f_p \tag{23}$$

Similarly for the driver pressure response:

$$P_{max, \, t \arg et} = P_{max,base} + P_{max,c} + P_{max,p} \tag{24}$$

We define each of these as respiratory effects $E_{c,p}$ where c denotes central effects and p denotes the peripheral chemoreceptor response. Using this definition, we denote the evolution of this response to be a function of arterial carbon dioxide partial pressure $P_{CO_2}(t)$ changes against a normal homeostatic set point $P_{CO_2,set}$:

$$\frac{d\Delta E_c}{dt} = \left(\frac{1}{\tau_{E,c}}\right)\left(-\Delta E_c + g_{E,c}(P_{CO_2}(t) - P_{CO_2,set})\right) \tag{25}$$

Here the effects $E = \{f, P\}$ are respiratory driver frequency $f$ and amplitude $P$ with associated time constant $\tau_{E,c}$ and controller gain $g_{E,c}$. We then define the peripheral feedback in a similar way to Eq. 25. As previously the effector equation computes a response for driver frequency and amplitude and is defined as:

$$\frac{d\Delta E_p}{dt} = \left(\frac{1}{\tau_{E,p}}\right)\left(-\Delta E_p + g_{E,p}(f(t) - f_{set})\right) \tag{26}$$

Here, $f(t)$ denotes the firing rate of the response and is a nonlinear sigmoid function of CO2 and O2:

$$\frac{df}{dt} = \frac{1}{\tau_f}(-f(t) + \psi)$$

$$\psi = \frac{f_{max} + f_{min} \exp\left(\frac{P_{O_2} - P_{O_2,half}}{k_{O_2}}\right)}{1 + \exp\left(\frac{P_{O_2} - P_{O_2,half}}{k_{O_2}}\right)} \left[K \ln\left(\frac{P_{CO_2}}{P_{CO_2,set}}\right) + \gamma\right] \tag{27}$$

Here $P_{O_2,half}$ is the half max and $k_{O_2}$ is the slope of the sigmoid response. K and $\gamma$ are tuning parameters used to maintain a steady state response under normal conditions. Values for the parameters are mostly taken from prior studies[45], or are tuned to meet validation data under various patient hypoxic states due to different injuries. The central nervous system then takes the chemoreceptor effects and weights them to compute an efferent response. Opioids act against the central nervous system by binding to the mu receptor, buffering the system's ability to control hypoxic conditions. We represent this action by perturbing the afferent firing input as a function of the central nervous system modifier, being generated by the opioid concentration on the effect site:

$$A_f = \psi e^{(-3.5\nu)} + A_{set}\left(1 - e^{(-3.5\nu)}\right) \tag{28}$$

Here A is the afferent firing input, and A-set is the associated set point, nu represents the central nervous modifier generated by the fentanyl concentration at the effect site. Similarly, the CNS modifier influences the central input:

$$C_f = (P_{CO2} - P_{CO2,set})e^{-3.5\nu} \tag{29}$$

Where C denotes the central frequency input.

## Reporting summary

Further information on research design is available in the Nature Portfolio Reporting Summary linked to this article.

## Results
### Pharmacokinetic profiles

**Naloxone.** To test the proper dosing of naloxone needed to reverse extreme overdoses involving fentanyl, we first validate the nasal model of naloxone. We collected data for 2, 4, 8, and 16 mg doses of naloxone and their associated venous plasma concentrations. The 2 and 4 mg naloxone dose concentration data was collected from[40]. Their study randomized 38,

**Table 1 | Root mean square error of simulated naloxone kinetics in the body after nasal administration**

| Naloxone Dose (mg) | Root Mean Square Error | As a percentage of peak concentration |
|---|---|---|
| 2 | 0.324 | 12 |
| 4 | 0.806 | 14 |
| 8 | 1.08 | 11 |
| 16 | 2.36 | 17 |

Validation data was used to fit the kinetic model for the 2 mg dose. The error term is an average vertical distance between the validation data and simulated data. The percentage is computed by scaling the error term by the maximum naloxone concentration value.

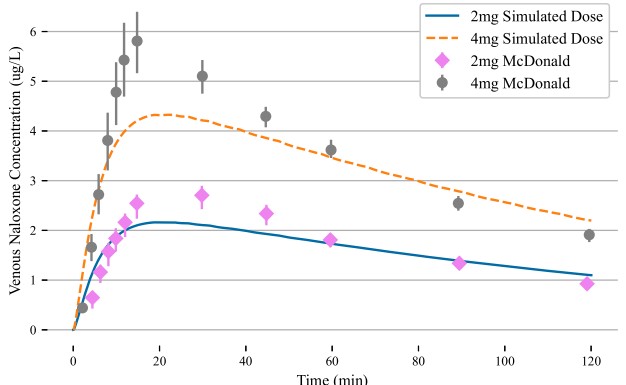

**Fig. 2 | 2-hour simulation of nasal naloxone dose for 4 (orange dashed) and 2 (blue) mg doses.** The peak venous concentration and long trend dynamics follow experimental data well, with minor peak concentration differences for each dose. Error bars correspond to one standard deviation of experimental data.

healthy male and female patients. With 19 blood samples taken over a 2-hour study period. We do note that the study population trended male, consisting of 71% of the study participants. Standard deviations were reported by the authors and that data was extracted for this comparative study. The experimental study also collected data for the following administration routes: intramuscular, intravenous, and a 1 mg intranasal dose. These results were omitted for this modeling study.

By hypothesizing that the naloxone required to resuscitate a patient who has been given, an extreme dose of fentanyl would need repeated, cumulative nasal administrations of a rescue drug, we investigated the pharmacokinetics of naloxone doses above the average dose administered by a single Narcan nasal spray. To this end, we collected 8 mg and 16 mg naloxone nasal dose data from[48] to test the model against. The Mundin study enrolled 12 subjects, 5 male and 7 female, with varying age, weight, and height. The naloxone nasal dose was formulated in the lab from a powdered dose mixed with sodium citrate and sodium chloride and atomized in a metered spray device. Blood was sampled at 10 distinct points throughout the six-hour study period with standard deviations computed for the population.

The nasal naloxone model presented here was fit directly to the 2 mg nasal dose (training) with the other doses presented here (4, 8, and 16 mg) used as validation (test). The fitting procedure was an iterative alteration of the kinetic parameters of the nasal deposition model to obtain an optimized plasma concentration. No kinetic parameters were changed after the fit to the 2 mg dose. As the naloxone circulated in the blood (represented here as a lumped parameter circuit system). Deposition and clearance were performed through computation of the partition coefficient for a given tissue compartment, Eq. 2. Physiochemical properties of naloxone were used to compute the partition coefficient, and none of these parameters were altered during the dose study, Supplementary Table 1.

To analyze the validity of using this model to study reversal and resuscitation efforts in the overdose patient, we first compared the validated data to the simulated data through the root mean square error (RSME). We note that root mean square error scales with the size of the naloxone dose (non-normalized), so to aid in understanding of the errors in the model we also compute an average percentage that corresponds to the percentage that the simulated data is different from the validation data (on average), Table 1. We also note that because the 8 mg and 16 mg data were collected from a different source, there were more data points available and a slightly different experimental protocol for the participants. This shouldn't impact the RSME as additional data points just contribute to the average distance incremental.

We note that the simulated data matches the rate of onset and long-term dynamics quite well, with much of the error being generated in the peak concentration profiles, Supplementary Fig. 2 and Fig. 2. The 16 mg dose displays the largest deviation from experimental data with a root mean square error of 2.36, computing to an average of 17% error between simulation and experimental data. Most of this error is generated during a overshooting of the peak reported naloxone concentrations. We do note that the 16 mg experiment displays the largest reported standard deviations when compared to the other simulations reported. In general, peak profiles were underestimated in the model, with all simulated data staying under a 20% deviation to experimental data, Tabel 1.

**Fentanyl**. We include a discussion on the pharmacokinetic profile of fentanyl administration, using limited available data. We note that there are no human studies available at the doses equivalent to illicit use that we consider late in this paper (relevant for street use and overdose amounts). Generally, these doses are considered unethical for human clinical trials due to the large respiratory depression that they generate. We purposefully omit animal studies as equivalent dose comparative studies due to the distinctly different respiratory action present in animal studies[49].

We begin our study of the kinetic profile of fentanyl by noting that the perfusion-limited diffusion partition coefficients were computed from the statistical analysis reported in 2.1.1. No parameter fitting was performed for these results, drug profiles are reported for the different administration pathways, concentrations, and total amounts as described in Table 2. In general, BioGears generic partition coefficient calculations do a good job modeling the peak concentrations in the infusion results, Fig. 3. Infusion administration peak fentanyl concentrations matched experimental data in both the Christrup and Ziesenitz studies, although the tail clearance was too rapid in the simulated data. This leads to discrepancies between the simulated fentanyl concentrations and experimental data of ~ 1 ug/L for infusion experiments. Bolus administration peak concentrations were generally greater in the simulated data but still within ~20 ug/L. The late-stage dynamics of the drug in the body in the simulated results remains elevated for almost all results. We note that these plasma concentrations of less than 1 ug/L, are not significant when considering severe respiratory depression. In general, the BioGears model clears the drug too rapidly to match the tails of the studies reported. We note that due to the more rapid clearance of the drug from the bloodstream in the simulations that we would expect a leftward shift of respiratory depression induced by fentanyl. Meaning that the onset and recovery of respiratory depression should occur more rapidly during the simulation. The consequences of this, as seen in the context of this project, would be that the first naloxone dose may be delayed in a real world setting and still provide sufficient resuscitation to the patient. The rate and timing of the oxygen saturation curve should be more rapid in the physiology model than in the patient.

From these concentrations, we aim to further validate the physiological response of the patient due to fentanyl administration. To that end we follow the experimental description from[50] to construct a mirrored simulated patient who are given a bolus dose of fentanyl (0.5 mg dose). We show that the respiratory depression taken as a function of an initial bolus dose of 0.5 mg is well validated when compared with experimental data, Fig. 4. We note that the time course of the respiratory depression follows experimental

**Table 2 | Details on experimental data used to validate the fentanyl patient response in the physiology engine**

| Route | Concentration | Total Dose (ug) | Reference |
|---|---|---|---|
| Infusion (5 minutes) | 50 ug/mL | 100 ug | 61 |
| Bolus (2-minute delay) | 50 ug/mL | 200 ug | 62 |
| Infusion (10 minutes) | 50 ug/mL | 5 ug/kg (350 ug for virtual patient) | 63,64 |
| Bolus | 50 ug/mL | 500 ug | 50,65 |

Data digitally extracted from figures. Data from[66] was omitted as the magnitude of the blood plasma curves indicated instrumentation error, or large deviations due to age of the study.

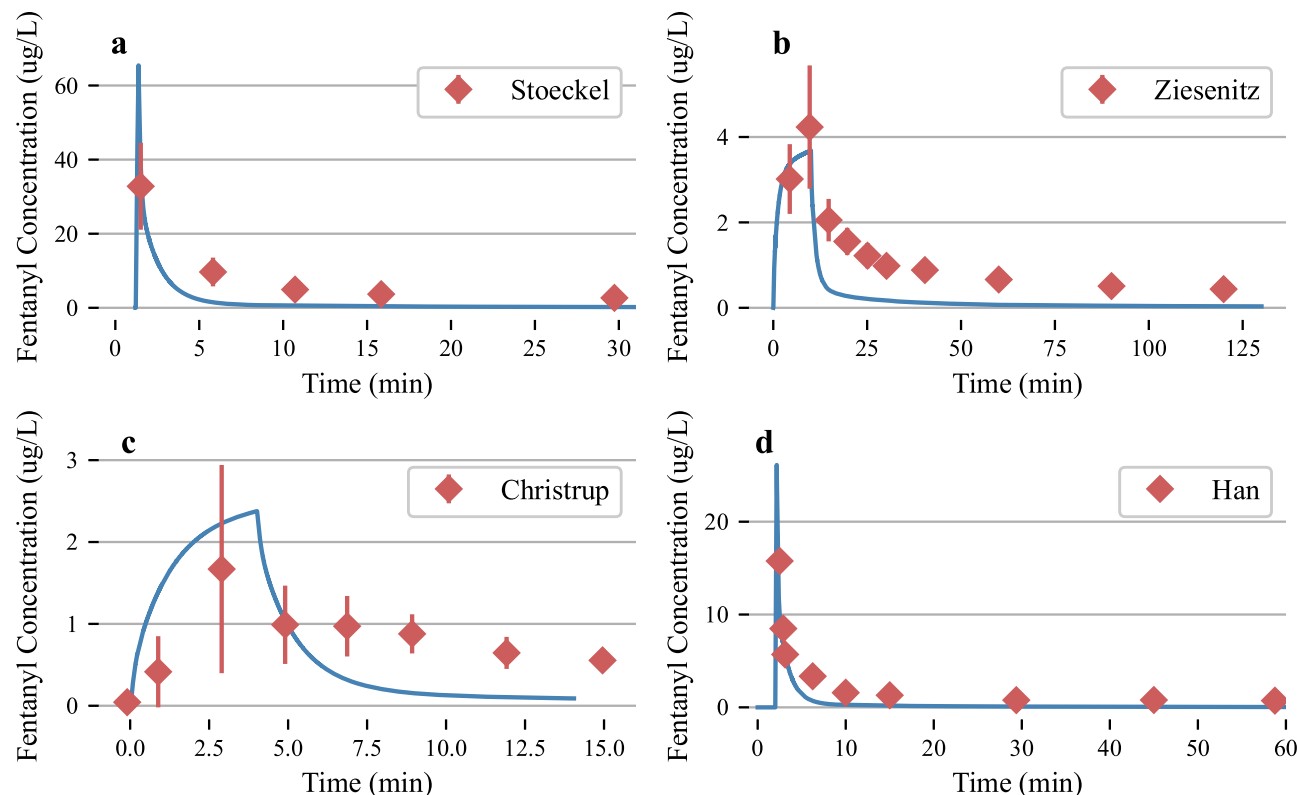

**Fig. 3 | Pharmacokinetic profiles for various routes of administration, concentrations, and total doses for experimental studies in humans.** Error bars correspond to one standard deviation of experimental data, deviations were not reported in the Han study. **a** and **d** simulate bolus dose administrations, equivalent to experimental studies. Subfigure **b** simulates an infusion for 5 minutes, and **c** simulates an infusion over 4 minutes.

studies, including the associated rate of depression and recovery. The overall magnitude of the pulmonary ventilation depression does not achieve the same levels reported in this study (a difference of ~1 L/min), but the data matches the simulations during the recovery period reported immediately after the maximum depression. The model respiratory recovery period also displays the oscillations seen in the experimental data where the chemoreceptors are influencing respiration after the initial fentanyl clearance. The overshoot in recovery seen in the model data is due to the nervous system parameterization which overcompensates due to the extreme hypoxia of the patient.

To further investigate the physiological impact of fentanyl on the patient, we report the oxygen partial pressure, as extracted from the aorta compartment in the simulated patient. We note that the minimum oxygen partial pressure is within the standard deviation of the experimental data but also note two issues with the simulated physiology: the minimum occurs too rapidly, and the recovery period occurs much quicker than the experimental data. Each of these issues relate to the oxygen blood/gas binding equations and diffusion equations that govern oxygen transport from the gas compartment of the alveoli into the pulmonary arteries. Validation of these models is area of future work.

To finalize the analysis of the physiological impact of fentanyl administration, we compute the CO2 response curve as a function of steady-state fentanyl concentration, Figs. 4 and 5. To perform this simulated experiment, we create a step infusion protocol that begins with a 10 ug/mL fentanyl concentration infused at a rate of 0.5 mL/min over the course of 10 minutes (an approximate steady-state plasma concentration). After 10 minutes we increment the concentration by 10 ug/mL up to a maximum of 60 ug/mL infused. During each 10-minute window we adjust the environmental substance composition to consist of 3% carbon dioxide, up from a standard composition of 0.04%. After the environmental inhaled air composition is changed, we let the patient respirate for five minutes, the end of which we revert to a traditional air composition. The respiratory impact of this increase in carbon dioxide exposure is evident in Fig. 5 with clear increase to respiration rate, pulmonary ventilation and end-tidal carbon dioxide. As the fentanyl concentration increases in the patient the response to these disturbances becomes diminished as the chemoreceptor feedback is buffered due to the fentanyl concentrations. The central nervous response as a function of fentanyl plasma concentration is also reported during this simulation. We then compute the $CO_2$ response by computing the slope of

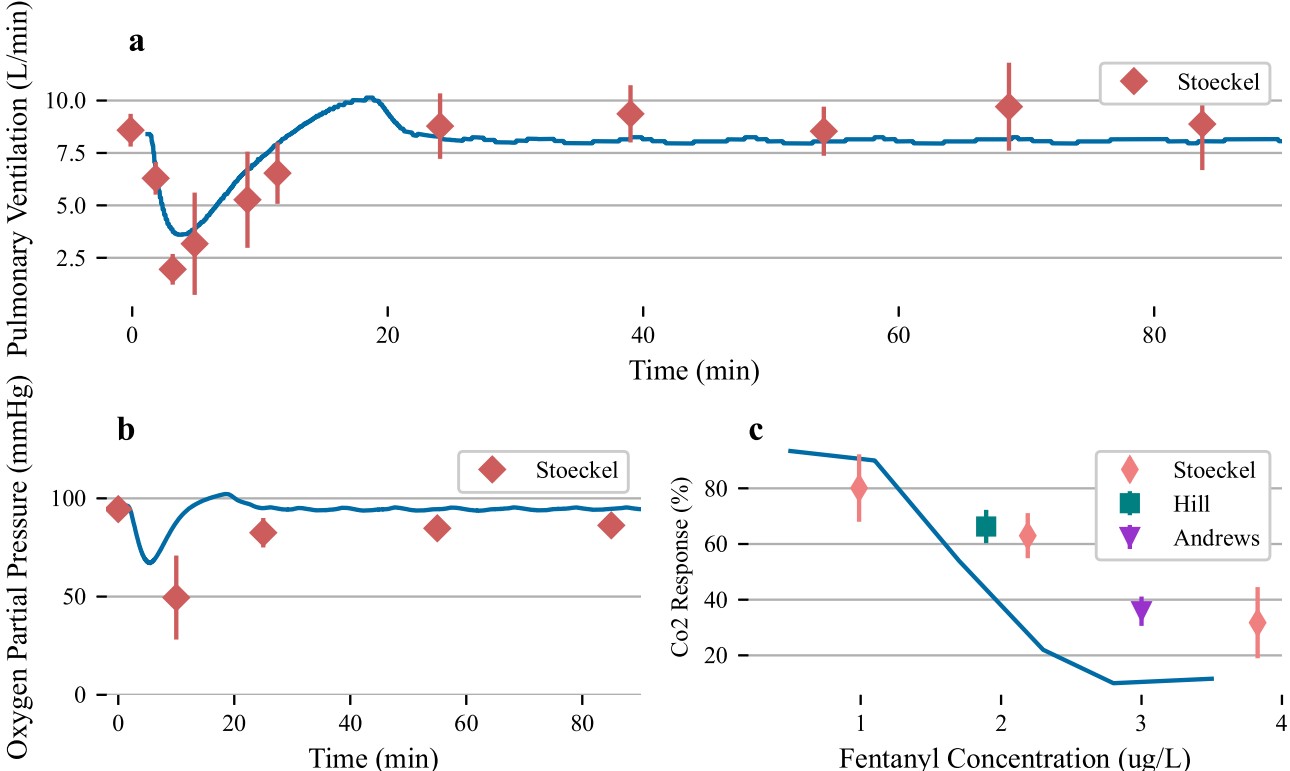

**Fig. 4 | CO₂ response for varying levels of Fentanyl bolus dose. a** Total pulmonary ventilation, **b** oxygen partial pressure for a single 0.5 mg bolus dose of fentanyl. Subfigure **c** displays the $CO_2$ response, computed for varying levels of steady-state fentanyl concentrations. Respiratory depression as a function of the fentanyl administration tracks well with the experimental data from[50]. Error bars represent one standard deviation from the mean of the data.

pulmonary ventilation and end-tidal carbon dioxide perturbations during each ten-minute window. The $CO_2$ response curve is reported against various experimental data points in Fig. 4.

The $CO_2$ response curve has a logistic shape and matches experimental data at lower fentanyl plasma concentrations. The middle slope of the logistic curve is sharper than the reported experimental data. We note a few differences between the experimental protocol between the reported data and the simulated data presented here. We were not able to completely match the $CO_2$ changes reported in the literature and instead leveraged the BioGears built-in environmental substance composition to change the $CO_2$ concentration that the simulated patient inhales. In general, we note that the curve has the right inherent shape as a function of fentanyl concentration but is shifted to the left of the experimental data. This could be due to the higher peak fentanyl concentrations for all pharmacokinetic experiments performed for this study, Fig. 3. As the peak concentrations are above experimental values, we may assume that the central nervous response in the physiological simulation is also more pronounced (more rapid onset) than the one seen in an actual human population. Ultimately the model captures, qualitatively, some of the major features of respiratory depression due to fentanyl administration. Validation of the blood/gas and diffusion models will be included in future work.

### Naloxone treatment analysis

To begin testing the naloxone administration amount needed for fentanyl overdose reversal we first created an algorithmic scenario centered around the patient's oxygen saturation ($SpO_2$). Assuming first responders would be able to access this piece of data, we built a repeat administration scenario to properly recover the $SpO_2$ over the length of the scenario. To begin the scenario, fentanyl was administered, followed by a two-minute waiting period. To assess the reversal dosing requirements, we incremented the initial fentanyl dose from 0.2 mg up to 1.9 mg, generally considered a very high dose for recreational use. We note that if the patient did not receive any

nasal naloxone dosing, the patient would die for doses of fentanyl over 1.5 mg.

To simulate the transit time of the emergency care provider we waited 2 minutes post opioid dose before administering naloxone. We note that this is an extremely rapid time to scene for first responders but wanted to consider a rapid patient response. For example, the average time to scene in King County Washington is 5.4 minutes[51].

Once the two-minute wait period has expired we administered an initial 2 mg rescue dose of naloxone, indicated by the purple line, Supplementary Figs. 3-6. After the initial rescue does was administered, we continuously monitored the patient's $SpO_2$ and at each 50 second increment, evaluated whether it was above a certain threshold value. We chose a value of 85% as a level of concern clinically. We note that generally a value below 90% would be considered hypoxic but did not want to overcompensate our dosing amounts by administering naloxone in the latter stages of the patient recovery. After 10 minutes of oxygen saturation over 85% we consider the patient rescued, no longer requiring more naloxone.

More complex goal directed therapies are possibly within the physiology engine, but we wanted to only understand the naloxone administration requirements as the sole rescue intervention. Other resuscitative interventions could be considered in the future, such as ventilation, bag valve mask, or supplemental oxygen. Additionally, the resuscitation scenario could have longer observation periods in between repeat does, which may alter the data presented in this manuscript. We note that the end-of-life thresholds for the physiology engine are rather crude and would need to be updated to firmly observe death in a patient from asphyxiation. All data was generated from a single C + + program using the BioGears engine[22]. The HowTo example file is open source in the core repository on GitHub[52].

**Respiratory Function.** Respiratory depression due to fentanyl administration is produced through the binding of the mu-opioid receptors at specific sites in the central nervous system. We simulate this action by

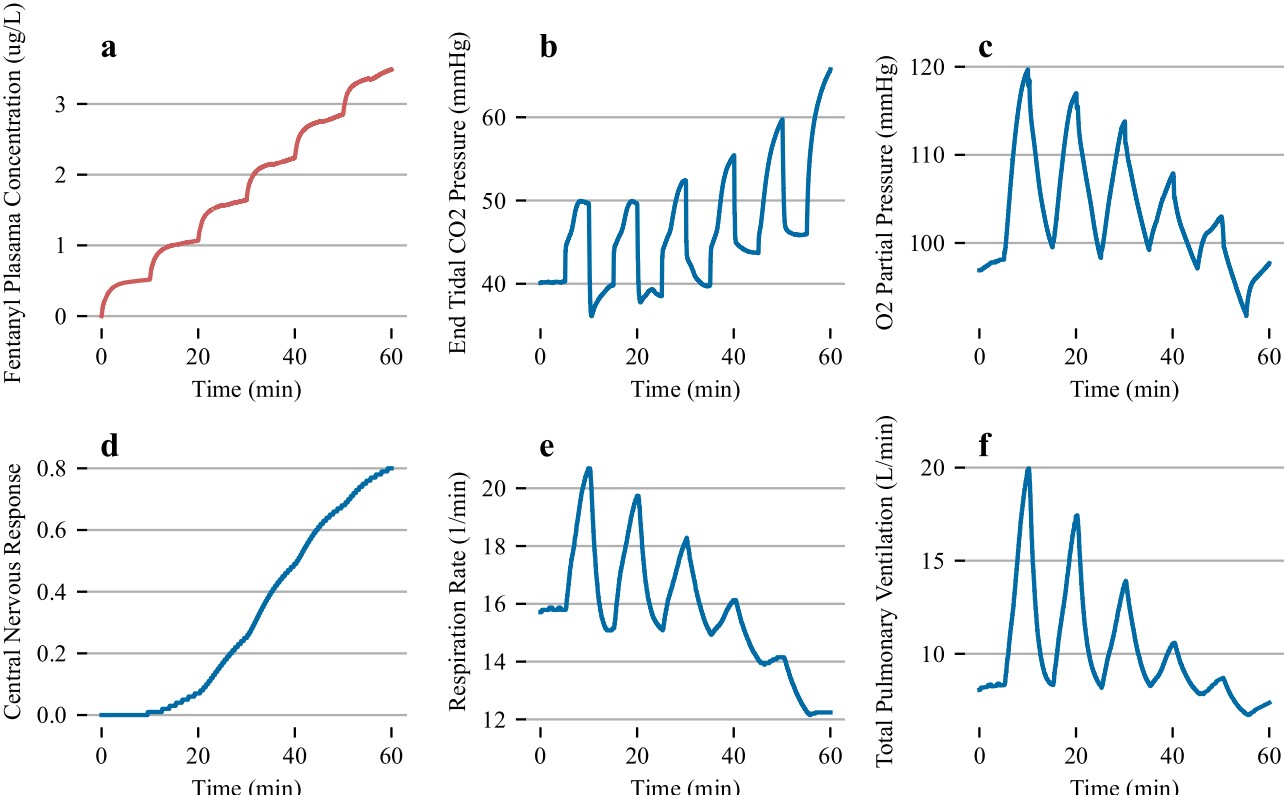

**Fig. 5 | Simulated testing of the co2 response as a function of fentanyl steady-state concentration in the blood.** Step infusions were administered to the simulated patient, and after five minutes of steady-state plasma concentrations, subfigure **a** displays the fentanyl plasma step concentrations, **b** displays the end-tidal $CO_2$, **c** displays the $O_2$ partial pressure, **d** displays the central nervous response, **e** and **f** highlight the respiration rate and total pulmonary ventilation over the duration of the simulated experiment. We altered the inhaled substance composition to consist of ~ 10% $CO_2$ for 5 minutes. The environmental inhaled substance composition was returned to normal before each fentanyl increment.

reducing the nervous system function in the physiology engine as a function of fentanyl blood concentration. This function in turn, drives the respiratory driver in the physiology engine. Central nervous depression directly depresses the respiratory cadence of the patient, Eqs. 28 and 29, thus inducing a hypoxic state in the patient, Fig. 6. Respiratory rate is properly depressed as a function of the initial dose of the patient, with mu-opioid receptor saturation incurred at doses above 1.5 mg. Number of breaths per minute in the patient drop to 2 from a baseline rate of approximately 16. Oxygen saturation in the physiology engine is computed as a function of blood-gas binding and oxygen diffusion through the alveolar compartment. Because the respiration rate is so dramatically reduced, oxygen saturation rates drop over the scenario duration. Recovery for large doses over 1 mg takes much longer than for lower doses of initial opioid administration. Oscillations seen in the oxygen saturation curves are due to respiratory function and circulatory oscillations seen in the models, as well as delayed interaction with the nervous system model. We note that a common patient response to opioids is irregular breath rhythm and tidal volumes[53–55]. The model presented here does show some minor variability in the respiration rate, although not consistent with the pronounced variability seen in many patients. The tidal volume, Supplementary Fig. 3, does display much more erratic variability, possibly due to the pronounced role of the peripheral nervous system response and associated changes in driver pressure during the simulation. Future work could potentially focus on modeling the coupling of the two primary oscillators present that regulate respiration frequency: pre-Botzinger complex and the nearby retro-trapezoid and parafacial respiratory group[56]. There is evidence to show fentanyl acts on only one of these, causing disruptions in the rhythm generated by the dynamic coupling of these oscillators. Finer representation of these neurological actuators and the influence of opioid receptor binding would be an excellent iterative advancement of the models presented here.

**Nervous system function.** The nervous system in the physiology engine is modeled to provide feedback to the body during times of perturbations from normal homeostatic physiological values. These types of perturbations can occur globally, oxygen partial pressure changes and blood pressure changes, or locally, drops in renal artery perfusion. Local changes are handled through autoregulatory relationships that work to preserve the perfusion to a given region, such as the brain or kidney. During an opioid overdose the patient oxygen partial pressure is perturbed in a significant way, leading to hypoxia. The acting mechanism by which is occurs is through fentanyl depressing the central nervous system. This depression leads to a decreased frequency and depth of respiratory function. The peripheral nervous system is allowed to respond accordingly, and the difference between the two responses can be seen in Supplementary Fig. 6. The central frequency can properly respond to the respiratory depression and associated reduction in oxygen partial pressure in the blood after the naloxone can properly reverse the opioid central nervous system binding.

Tidal volume properly responds to the reduction in oxygen in the blood by increasing the amount of volume per breath. This is achieved through increases in the peripheral pressure changes, which alter the respiratory diver pressure. We report this pressure as a cmH2O and note that it may be scaled to mmHg using a conversion constant of 0.735559. This pressure difference creates a larger negative potential for the lungs to fill during a single cycle. We note that the tidal volume generally should decrease for this amount of fentanyl administration with other research suggesting decreased volumes during increased opioid exposure[55]. Other opioids have been shown to not decrease tidal volumes[57] leading to unique respiratory

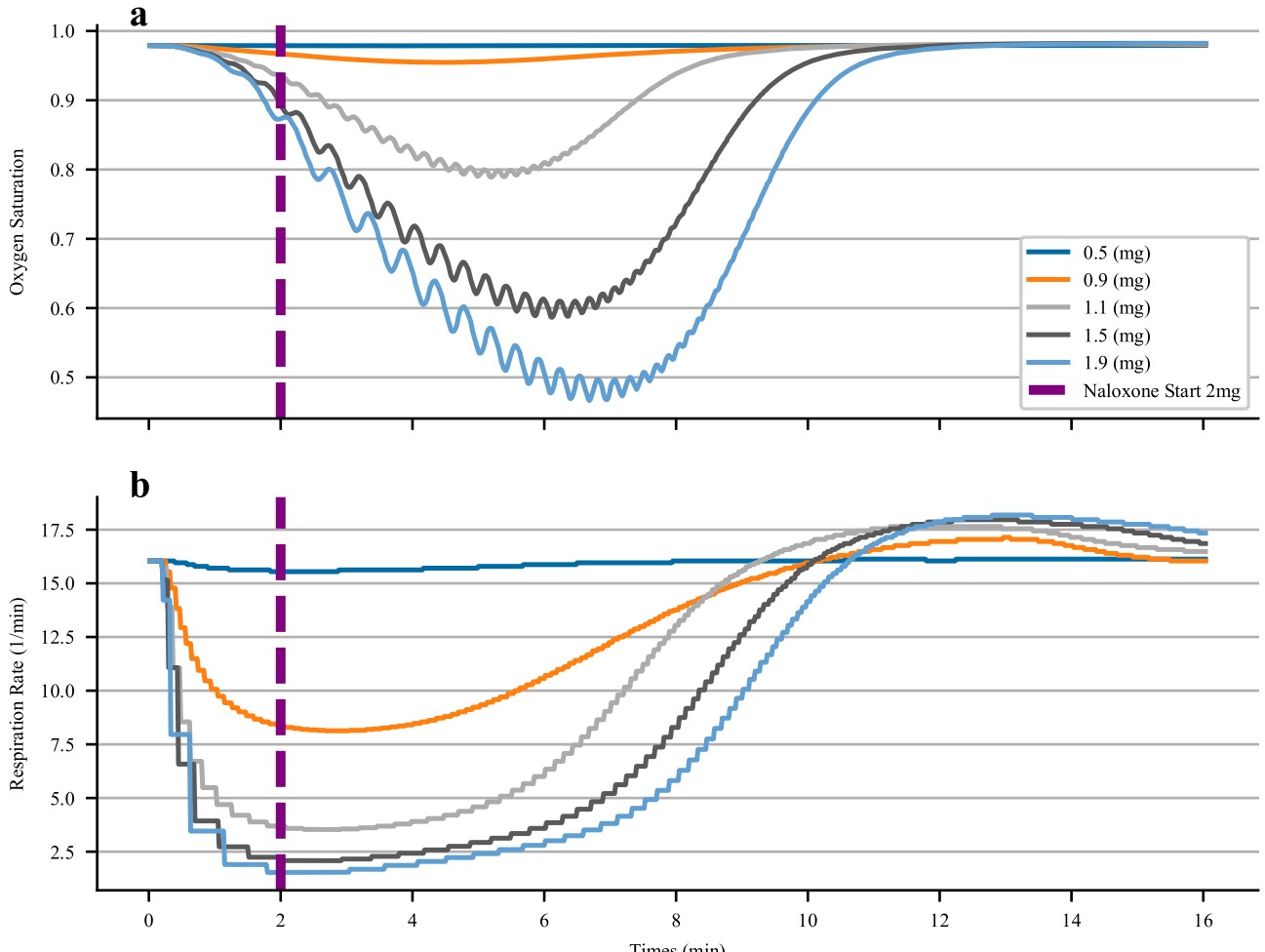

**Fig. 6 | The primary cause and effect of fentanyl induced hypoxia are plotted here for varying levels of initial fentanyl bolus doses.** Subfigure **a** displays oxygen saturation curves over different initial fentanyl doses, and **b** highlights respiration rate over those same fentanyl doses. The purple line indicates the initial naloxone dose administered during the rescue scenario.

responses depending on the type of opiate the patient has been exposed to. The qualitative issues shown here can be mitigated in future work by including a term to the drug file that includes binding affinity for specific receptors, leading to a much more robust opioid model.

**Total naloxone administration.** Despite the prevalence of naloxone accompanying emergency response providers, the rate of synthetic opioid overdoses and associated mortality is not decreasing. As stated above the use of multiple doses to reverse synthetic opiates is increasing in some localities[13], although other studies show that in specific counties this trend is not mirroring the rise in opiate use[14]. Generally, the use of naloxone has mirrored the rise in mortality in the age-adjusted population[8]. This data points to a potent intervention tool that is potentially not being used in sufficient amounts to address the rise of synthetic opioids in the population.

To provide some understanding of the dosing requirements of naloxone to reverse extreme fentanyl exposure, we sought to provide a computational model that could start to answer that question. Following the algorithm in Fig. 7 we summed up the total naloxone reversal dose for a specific overdose event, Supplementary Fig. 7. For larger fentanyl doses, there seems to be an inflection point in the data, this occurs around the 0.9 mg dose, and continues a rapid, nonlinear increasing response up to the highest range of fentanyl. Although we understand the initial dose of fentanyl being administered the amount of naloxone needed to reverse the patient's respiratory depression is only a function of the patient's oxygen

saturation. We are not implying that rescue EMT's would be able to know the initial fentanyl dose a priori. We note that a standard dose of Narcan is 4 mg, even though the standard dose we used for this analysis was 2 mg, although some doses currently deployed are of the 2 mg formulation. We note that for a lower dose of fentanyl, the standard dose of 2 mg tends to reverse the respiratory depression sufficiently, with no relapse and with sufficient onset to deter the hypoxia seen the in the patient.

Noting that there existed an inflection point in the data, we sought to understand the general statistics between three distinct bins of fentanyl dose. We collected 6 doses and computed their mean, median and standard deviation, Fig. 8. From these statistics we computed the *p*-value to compare whether each bin of fentanyl doses was statistically significant from the prior bin, Supplementary Table 4. The sample size for the test was equal to the size of the discretized bin, equal to 6 for this study. For each comparison we note the statistically significant dose of naloxone required to reverse the opioid-induced respiratory depression, *p*-value of 0.0006 between the first and second bin and 0.0018 between the second and third bins, confirming our hypothesis that the dose requirements for naloxone, via nasal administration, were insufficient for high levels of fentanyl for this specific modeling study.

## Discussion

The opiate overdose death rate has surpassed motor vehicle collisions as the leading cause of death for ages 18–64 and for every death there are 6.4-8.4 non-fatal overdoses[58]. The number of synthetic opiates-related deaths is

**Fig. 7 | Simulated protocol for reversal resuscitation for an opioid overdose patient.** Green denotes the start of the rescue period. Red boxes indicate decision/administration points, with blue boxes denoting observation or translation points.

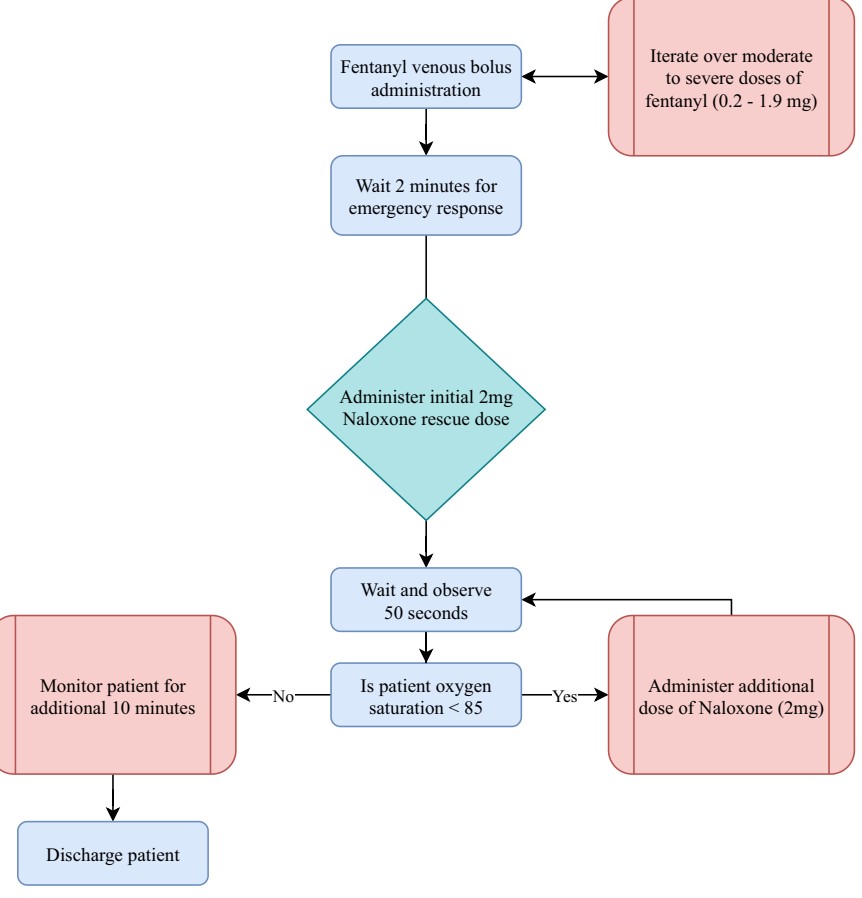

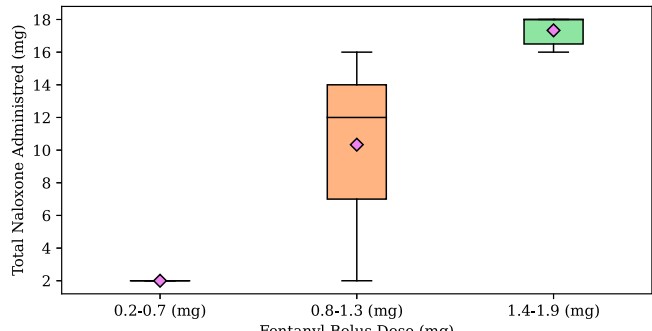

**Fig. 8 | We group three fentanyl doses from analyzing the total administration amounts of the simulated experiment and compute the mean (pink diamonds) and standard deviation of the total naloxone dose.** The line in the middle of the boxes indicates the median of the data. Each box plot quantifies the non-linear response of the required naloxone dose as a function of fentanyl dose. $N = 6$ for each sample grouping.

administered to reverse the respiratory depression effects, up to 8x the standard dosing given by EMS. There is evidence that the timing of this administration, as seen in these models, also plays a role. One of the reasons synthetic fentanyl leads to such a frequency of overdose is the time to onset of respiratory depression. This type of rapid respiratory distress can be seen in the figures generated during this simulated treatment scenario. The peak concentration of opiate is higher and reached faster than seen previously in heroin overdose, 2 compared to 30 minutes[59]. This degree of opiate intoxication leads to faster, more profound hypoxemia due to a decreased respiratory drive with a refractory recovery period compared to previously seen morphine equivalents. This leaves urgency for first responders regarding how many doses of naloxone may be required to adequately reverse an overdose.

Combining the physiologic effects of fentanyl with the timing of peak concentrations of the higher doses of naloxone can guide timing of monitoring after administration. Typically, if there is concern for severe opiate overdose, a patient should be transported to the nearest capable emergency department for monitoring and any further treatment that should be needed. In cases of refusal of transport, which occurs in approximately 40% of naloxone administration cases, patients requiring multiple doses of naloxone for reversal would imply higher serum concentrations of opiate and therefore a slower response for a sustained safer respiratory rate and oxygen saturation[60]. A higher dose of the opiates, as seen in these models, would also theoretically lead to a slower resolution of hypoxemia and bradypnea even with a normal concentration of naloxone dosing. Understanding these physiologic derangements can offer guidance to first responders to approximate or anticipated dosage with suspected fentanyl overdose amounts in the patient and provide information for the time interval for recovery that should be anticipated.

We note that the complete resuscitation picture is not present in this study. Generally, ventilation would be required for the doses of fentanyl being given to the patient in these simulations. We omit an analysis of

climbing and despite adequate supply of naloxone provided to first response providers, this death rate continues to climb[9,58]. There are misconceptions and a general lack of knowledge surrounding the physiology of synthetic-derived opiates, which may be leading to under and ineffective utilization of naloxone at doses adequate to reverse the extremely high serum concentrations that these synthetic opiates reach at an impossibly short time interval, making them particularly lethal. These models quantify the dosing curves of the very high doses of fentanyl and the associated nasal naloxone treatment that are achieved in the blood, which will be instrumental in understanding appropriate reversal dosing.

These models demonstrate previously held understandings that in higher doses of synthetic opiates, higher doses of naloxone should be

**Article**

ventilation but will revisit this subject in future research. Additional future work should include respiratory collapse due to extreme overdose amounts. Such a collapse would change the compliance in the lungs to properly model the wooden chest syndrome and properly interact with the ventilation models to quantify the resuscitation needs for such patients.

The advantages of an accurate model of the pharmacokinetics of fentanyl, naloxone, and associated respiratory depression are the ability to design and investigate repeat dosing scenarios in a safe environment. Limitations exist in animal models related to the translation of human to animal dosing equivalents, as well as obvious anatomical differences that may impact investigations regarding the route of naloxone administration. Human models are limited to extremely low doses of opioids to ensure the safety of the participants of the study. A computational platform provides a safe, repeatable environment to design investigative studies as a first pass at obtaining information regarding changes in requirements of medical care. These investigations may also be a platform to inform future clinical trials and experimental studies that focus on naloxone plasma concentrations as a function of hypoxemia of the patient. How does naloxone interact with other types of opioids in the nervous system and what rate does it compete with these synthetic opioids as an agonist for selective nervous system binding? The ability for the BioGears physiology platform to be extended to include a larger collection of synthetic opioids could open the scope of the results reported here. The physiochemical properties detailed in Supplementary Fig. 8 are required to model any new drug in the platform. This experimental data then informs the clearance, deposition, and influence on the patient for any dose (including different routes of administration) that the user defines.

Many flaws exist in the current model that need to be addressed in future iterations. These include nasal anatomical structure refinements, fentanyl overdose clearance and binding changes, and nervous system model changes to consider the different binding pathways that the opioid may prefer depending on the type and amount administered. Including multiple receptor sites and associated impact on nervous function rather than just the mu pathway is one way to expand the current model. Clear issues regarding tidal volume compensation due to hypoxemia, seen at low doses of fentanyl, need to be addressed for higher levels of the opioid.

In conclusion, we present a model of the diffusion and transport of naloxone into the blood-stream from a nasal administration pathway, coupled to a venous administration pathway of fentanyl. For all the drugs, we compute their perfusion limited diffusion into the tissue spaces during transport, leading to clearance in the body. We demonstrate that the model of naloxone for high levels of administration validates well against experimental data. We couple the results of the plasma concentration of fentanyl to a pharmacodynamic model of the respiratory depression and mu receptor binding influence in a central nervous system model. We demonstrate that these coupled models can induce severe respiratory depression and associated hypoxemia because of high levels of fentanyl administration. In addition, we capture the reversal of the patient physiology due to the naloxone in the body due to a competitive antagonist drug-drug interaction model. We note $p$-values less than 0.05, Supplementary Table 4, when comparing the initial fentanyl dose and the required naloxone reversal dose. We hope that this modeling result will induce more discussion regarding naloxone dosing requirements when it comes to synthetic opioid overdose for first responders in the field.

## Data availability
The datasets generated and analyzed for this study (source data) can be found in the repository of the data as well as the python notebooks used to generate all images and figures https://github.com/ajbaird/PaperMaterials/tree/main/NaloxoneFentanyl2023.

## Code availability
The software used to generate the provided data is also available free and open source in the BioGears core repository https://github.com/

BioGearsEngine/core, specifically the HowTo-NaloxoneNasal.cpp file was used to generate all the data in this paper[52].

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

## Acknowledgements

We'd like to acknowledge the support and guidance of Hugh Connacher, Harvey Magee, Dr. Brett Talbot, and Geoff Miller who all provided excellent guidance during the original development of the BioGears project. We'd also like to acknowledge critical members of the BioGears development team over the years: Matthew McDaniel, Jenn Carter, Jeff Webb, Aaron Bray, and Rachel Clipp. We also thank Applied Research Associates for supporting this open-source platform and continuing to usher its software development and architecture to aid its use in physiology modeling and research. No external funding was used for in this study.

## Author contributions

Conceptualization was done by A.B., S.A.W., R.D, data curation by A.B., formal analysis by A.B. and N.T., investigation was performed by A.B., N.T., and S.A.W, methodology developed by A.B., N.T. and R.D., project administration performed by A.B., S.A.W., E.B., resources provided by A.B., and S.A.W., software developed by A.B., S.A.W. and N.T., validation performed by A.B., and E.B., visualizations developed by A.B., primary and subsequent revision drafts written by A.B. and E.B., editing and review by A.B., S.A.W., E.B., N.T. and R.D.

## Competing interests

The authors declare no competing interests.
