## [Peer review file · Communications Medicine]

Whole body physiology model to simulate respiratory depression of fentanyl and associated naloxone reversalReviewers' comments:

Reviewer #1 (Remarks to the Author):

This is a very content-rich paper. It can be divided into two major parts: 1) development of a physiologically detailed (and hence technically sophisticated) model to simulate human responses to opioid overdose and naloxone rescue; 2) use the model to simulate various overdose scenarios and provide insight into optimal naloxone dosage and dosing strategies (e.g., repeat dosing).

Both parts are ambitious and almost warrant standalone papers. For the first part, technical details are given as various mathematical equations. While this provides very valuable resources for fellow modelers to develop/modify/expand their own models, ultimately the strength of any model, including this one in the manuscript, relies on the capability of reproducing real (in this case clinical) data. On that front this paper provides a comparison of their model simulation to 1) plasma profile of naloxone after nasal spray (Fig 3&4); 2) plasma profile of fentanyl after various dosing (Fig 5); 3) minute ventilation profile after fentanyl IV bolus injection (Fig 6). While the simulation is not perfectly in line with experimental data, this at least gives us confidence that the general trajectory of the simulated profile is not far from human data, and it points out ways to further improve the model.

Unfortunately, the comparison to clinical data stopped here, with some important omissions. For example, blood oxygen level, which is the main endpoint used for the second part when judging naloxone efficacy, has no such comparisons, despite the fact that some papers referenced by the authors do contain PaO₂ data after opioids administration (PaO₂ can be converted to oxygen saturation, which is used by the authors later). Also, some of those papers contain the change of CO₂ response slope after opioids administration, which is important clinical technique (and datasets) to quantify opioids effects on respiration.

Now moving on the second part. Assuming the model has certain credibility (but see the point of lack of clinical comparison above), the authors simulated applying different doses of naloxone to rescue patients with different fentanyl overdose, and saw a need for higher naloxone dose for higher fentanyl dose. While this makes total sense, it is unclear which of the simulated scenarios are closer to "real" situations in the community. What are "typical" dosage for fentanyl overdose? What if the naloxone dosing time is varied? What if a virtual patient took a long time to recover back to 85% oxygen saturation (authors' operational definition of rescue), so long that he/she had permanent tissue damage (so no "real" rescue)?

To be fair, asking the authors to add so many things (both part 1 and part 2) is a lot. On the other hand, without these addition, neither the first nor the second part is a complete investigation. The authors could consider splitting the paper into two. Alternatively, the authors could focus on the first part, provide a full comparison to various clinical data, and show part 2 as a demonstration of model utility (and acknowledging the uncertainty regarding the real community situations).

Reviewer #2 (Remarks to the Author):

Interesting model of OIRD and naloxone reversal. Some items that need serious contemplation and adaptation:

1. The fentanyl model seems not appropriate, please address, particularly since the PK mismatch is appreciable - Fig 5-. Why not use a much simpler PK naloxone model with one or more absorption compartments??
2. You mention "markers" in the abstract, but do not mention them again. What is meant by markers

and why use this statement once?

3. LM391 Fit to 2 mg dose: how and which parameter values had to be adjusted? First present the fit and later the validation results with other doses. Does Figure 4 for dose 2 mg represent the fit? The gray and purple drawings should be explained in the legend.

4. LM 408 A deviation threshold of 20% is mentioned, and at LM 433 a 50% threshold - these seem arbitrary.

5. Figure 5: It would seem the mismatch between the fentanyl model and available data is quite large. What are the consequences for the conclusions of the paper?

6. Figures 8-12 Although the 2-min naloxone start is marked, is it correct that no naloxone is given?

7. Figure 9: Why is there a sudden decrease in cardiac output after a period of almost linear increase? What do the spikes in the curves mean?

8. Figure 12: cm H₂O as unit?

9. LM 792: What does statistical significance mean here?

10. Fig. 13 Is there an indication of variability. Your suggestion of efficacy of 2 mg naloxone does not fit reality, it is much less effective than suggested by fig 13.

The main question that arises is whether the results of the simulated modeling is realistic and may be applied to humans?

Reviewer comment response:

First want to thank the authors for their time and effort spent reviewing this article. The feedback was extensive and aided in the resubmission. The revisions presented here enhance the results dramatically, notably the, now updated, fentanyl section provides critical context for the reversal results presented later in the manuscript.

This is a very big paper with a lot of different results, but we felt that presenting a large comprehensive paper was better, perhaps, than splitting up results into multiple manuscripts. This way the readers can have all the details required to replicate this work and extend it, if possible, in one paper. Another notable addition is a large supplementary volume that provided extra details on the models and algorithms in the BioGears physiology engine. It is an excellent resource and overview of the general fluid transport models and will hopefully help people understand that these results aren't just a pkpd model but include multiple interconnected systems working together and in feedback with one another.

We note (as did the reviewers) there are still a way to go to fully validate the engine and its response to various trauma. We agree completely and will focus future research on sensitivity analysis of the various models, global optimization of the parameter space for these same models, further validate on real patient data, and report on the results. We fully support open science and all the effort that it takes to create truly "open source" scientific tools. We provide all the data used to create the results, C++ API file used to generate them, and the core physiology engine needed for that recreation all free and open source. We hope to continue to push code and updates to BioGears to be a flexible platform for whole-body physiology modeling and research.

Reviewers' comments:

Reviewer #1 (Remarks to the Author):

This is a very content-rich paper. It can be divided into two major parts: 1) development of a physiologically detailed (and hence technically sophisticated) model to simulate human responses to opioid overdose and naloxone rescue; 2) use the model to simulate various overdose scenarios and provide insight into optimal naloxone dosage and dosing strategies (e.g., repeat dosing).

Both parts are ambitious and almost warrant standalone papers. For the first part, technical details are given as various mathematical equations. While this provides very valuable resources for fellow modelers to develop/modify/expand their own models, ultimately the strength of any model, including this one in the manuscript, relies on the capability of reproducing real (in this case clinical) data. On that front this paper provides a comparison of their model simulation to 1) plasma profile of naloxone after nasal spray (Fig 3&4); 2) plasma profile of fentanyl after various dosing (Fig 5); 3) minute ventilation profile after fentanyl IV bolus injection (Fig 6). While the simulation is not perfectly in line with experimental data, this at least gives us confidence that the general trajectory of the simulated profile is not far from human data, and it points out ways to further improve the model.

Unfortunately, the comparison to clinical data stopped here, with some important omissions. For example, blood oxygen level, which is the main endpoint used for the second part when judging

naloxone efficacy, has no such comparisons, despite the fact that some papers referenced by the authors do contain PaO₂ data after opioids administration (PaO₂ can be converted to oxygen saturation, which is used by the authors later). Also, some of those papers contain the change of CO₂ response slope after opioids administration, which is important clinical technique (and datasets) to quantify opioids effects on respiration.

We agree with the reviewer and felt that the discussion regarding respiratory impact of fentanyl administration required a significant expansion. The justification for this expansion that the reviewer cites is shared by the authors and we have thus tried to include more details into this section of the results.

We've added one minor and one major experiment to test the model against existing experimental respiratory depression metrics. For the study on pulmonary ventilation, we also included a figure on the oxygen partial pressure. A new major contribution to this paper is investigating the CO₂ response as a function of fentanyl steady state concentrations. To complete this investigation, we've added a new data set and code to re-create the simulated experiment and have included a lengthy discussion on the results of this simulated experiment. Ultimately, there is still more to do to properly validate the blood/gas models in the physiology engine but this analysis was appropriate for this paper and is included in the main manuscript.

Now moving on the second part. Assuming the model has certain credibility (but see the point of lack of clinical comparison above), the authors simulated applying different doses of naloxone to rescue patients with different fentanyl overdose, and saw a need for higher naloxone dose for higher fentanyl dose. While this makes total sense, it is unclear which of the simulated scenarios are closer to "real" situations in the community. What are "typical" dosage for fentanyl overdose? What if the naloxone dosing time is varied? What if a virtual patient took a long time to recover back to 85% oxygen saturation (authors' operational definition of rescue), so long that he/she had permanent tissue damage (so no "real" rescue)?

We add a discussion about these what ifs to the results section of the paper

To be fair, asking the authors to add so many things (both part 1 and part 2) is a lot. On the other hand, without these addition, neither the first nor the second part is a complete investigation. The authors could consider splitting the paper into two. Alternatively, the authors could focus on the first part, provide a full comparison to various clinical data, and show part 2 as a demonstration of model utility (and acknowledging the uncertainty regarding the real community situations).

Ultimately, we have chosen the latter route for the edits of this manuscript. Adding a significant portion to the fentanyl results has given a much broader picture to the reader of the strengths and weaknesses of the model. Future work will target the "what ifs" presented here such as variable dose timing, changing the recovery criteria, and adding variability to the CNS response of the patient.

Reviewer #2 (Remarks to the Author):

Interesting model of OIRD and naloxone reversal. Some items that need serious contemplation and adaptation:

1. The fentanyl model seems not appropriate, please address, particularly since the PK mismatch is appreciable - Fig 5-. Why not use a much simpler PK naloxone model with one or more absorption compartments??

We've included a much more detailed analysis of the fentanyl PK characteristics in the results section and have updated the model language to properly justify the use of the three compartment model detailed in the manuscript, line 289 highlighted with additional citations.

2. You mention "markers" in the abstract, but do not mention them again. What is meant by markers and why use this statement once?

Abstract is updated with details on the "markers" mentioned.

3. LM391 Fit to 2 mg dose: how and which parameter values had to be adjusted? First present the fit and later the validation results with other doses. Does Figure 4 for dose 2 mg represent the fit? The gray and purple drawings should be explained in the legend.

Great question, the fit was an adjustment of the kinetic parameters of the nasal deposition model. The text has been update to include this. Legend included in the naloxone pk curves for clarity.

4. LM 408 A deviation threshold of 20% is mentioned, and at LM 433 a 50% threshold - these seem arbitrary.

We've re-written and updated the paragraph describing the error between simulated and experimental data.

5. Figure 5: It would seem the mismatch between the fentanyl model and available data is quite large. What are the consequences for the conclusions of the paper?

beginning in line 432 we provide context for the fentanyl pk curve differences in the context of the given study and hope that this helps the reader understand some of the issues with the computational model when compared to experimental data.

6. Figures 8-12 Although the 2-min naloxone start is marked, is it correct that no naloxone is given? sorry if this wasn't clear, I've updated the language on line 516 to be more clear and figure 9-13

7. Figure 9: Why is there a sudden decrease in cardiac output after a period of almost linear increase? What do the spikes in the curves mean?

Beginning at line 625 we've added an updated description for figure 10, to help describe the dynamics occurring.

8. Figure 12: cm H2O as unit?

We report the unit in centimeter water but included language in the text to perform the conversion to mmhg if the reader would like.

9. LM 792: What does statistical significance mean here?

This has been clarified to explicitly call out the p-values for the computational experiments being performed for this paper (highlighted).

10. Fig. 13 Is there an indication of variability. Your suggestion of efficacy of 2 mg naloxone does not fit reality, it is much less effective than suggested by fig 13.

For this simulation, we only tested on a single patient, so variability wasn't present in the results. Investigations concerning populations of individuals (in a simulated setting) will be an area of future research. Patient variability in regards to respiratory depression and nervous system firing rates would be ideal stochastic parameters to introduce for that study but are not considered here. We note that 2mg naloxone dose is a standard rescue dose administer by first responders, as is a 4mg nasal spray. For these lower fentanyl values, this dose should be sufficient to reverse their symptoms, although other physiological consequences, which aren't considered in this study, such as reduced compliance of the lungs may provide complications to the resuscitation that aren't considered here with these small doses.

The main question that arises is whether the results of the simulated modeling is realistic and may be applied to humans?

We hope that these changes have given a clearer picture as to how this study should be interpreted. Ultimately it is a presentation of a whole-body physiological model of naloxone and fentanyl with some major limitations, discussed throughout the manuscript. It is only a first step towards creating a platform that could perform robust, validated, simulations of these patients (especially over populations). This manuscript is the first step in building a body of validity for the models. There is still a long way to go and we hope that we are honest in our assessment in the model performance and the real world impact of what is reported here. We sincerely thank the reviewer for their contributions, they have made the research much more robust.

REVIEWERS' COMMENTS:

Reviewer #1 (Remarks to the Author):

Thanks for revising the manuscript by expanding the first part of the manuscript and acknowledging the limitation of the second part. Now it gives a clear and full picture of the strengths and weaknesses of the model. I have no other concerns.

Reviewer #2 (Remarks to the Author):

I thank the authors for addressing most of my issues. I have no further issues.

Reviewer comment response:

First want to thank the authors for their time and effort spent reviewing this article. The feedback was extensive and aided in the resubmission. The revisions presented here enhance the results dramatically, notably the, now updated, fentanyl section provides critical context for the reversal results presented later in the manuscript.

This is a very big paper with a lot of different results, but we felt that presenting a large comprehensive paper was better, perhaps, than splitting up results into multiple manuscripts. This way the readers can have all the details required to replicate this work and extend it, if possible, in one paper. Another notable addition is a large supplementary volume that provided extra details on the models and algorithms in the BioGears physiology engine. It is an excellent resource and overview of the general fluid transport models and will hopefully help people understand that these results aren't just a pkpd model but include multiple interconnected systems working together and in feedback with one another.

We note (as did the reviewers) there are still a way to go to fully validate the engine and its response to various trauma. We agree completely and will focus future research on sensitivity analysis of the various models, global optimization of the parameter space for these same models, further validate on real patient data, and report on the results. We fully support open science and all the effort that it takes to create truly "open source" scientific tools. We provide all the data used to create the results, C++ API file used to generate them, and the core physiology engine needed for that recreation all free and open source. We hope to continue to push code and updates to BioGears to be a flexible platform for whole-body physiology modeling and research.

Reviewers' comments:

Reviewer #1 (Remarks to the Author):

This is a very content-rich paper. It can be divided into two major parts: 1) development of a physiologically detailed (and hence technically sophisticated) model to simulate human responses to opioid overdose and naloxone rescue; 2) use the model to simulate various overdose scenarios and provide insight into optimal naloxone dosage and dosing strategies (e.g., repeat dosing).

Both parts are ambitious and almost warrant standalone papers. For the first part, technical details are given as various mathematical equations. While this provides very valuable resources for fellow modelers to develop/modify/expand their own models, ultimately the strength of any model, including this one in the manuscript, relies on the capability of reproducing real (in this case clinical) data. On that front this paper provides a comparison of their model simulation to 1) plasma profile of naloxone after nasal spray (Fig 3&4); 2) plasma profile of fentanyl after various dosing (Fig 5); 3) minute ventilation profile after fentanyl IV bolus injection (Fig 6). While the simulation is not perfectly in line with experimental data, this at least gives us confidence that the general trajectory of the simulated profile is not far from human data, and it points out ways to further improve the model.

Unfortunately, the comparison to clinical data stopped here, with some important omissions. For example, blood oxygen level, which is the main endpoint used for the second part when judging

naloxone efficacy, has no such comparisons, despite the fact that some papers referenced by the authors do contain PaO₂ data after opioids administration (PaO₂ can be converted to oxygen saturation, which is used by the authors later). Also, some of those papers contain the change of CO₂ response slope after opioids administration, which is important clinical technique (and datasets) to quantify opioids effects on respiration.

We agree with the reviewer and felt that the discussion regarding respiratory impact of fentanyl administration required a significant expansion. The justification for this expansion that the reviewer cites is shared by the authors and we have thus tried to include more details into this section of the results.

We've added one minor and one major experiment to test the model against existing experimental respiratory depression metrics. For the study on pulmonary ventilation, we also included a figure on the oxygen partial pressure. A new major contribution to this paper is investigating the CO₂ response as a function of fentanyl steady state concentrations. To complete this investigation, we've added a new data set and code to re-create the simulated experiment and have included a lengthy discussion on the results of this simulated experiment. Ultimately, there is still more to do to properly validate the blood/gas models in the physiology engine but this analysis was appropriate for this paper and is included in the main manuscript.

Now moving on the second part. Assuming the model has certain credibility (but see the point of lack of clinical comparison above), the authors simulated applying different doses of naloxone to rescue patients with different fentanyl overdose, and saw a need for higher naloxone dose for higher fentanyl dose. While this makes total sense, it is unclear which of the simulated scenarios are closer to "real" situations in the community. What are "typical" dosage for fentanyl overdose? What if the naloxone dosing time is varied? What if a virtual patient took a long time to recover back to 85% oxygen saturation (authors' operational definition of rescue), so long that he/she had permanent tissue damage (so no "real" rescue)?

We add a discussion about these what ifs to the results section of the paper

To be fair, asking the authors to add so many things (both part 1 and part 2) is a lot. On the other hand, without these addition, neither the first nor the second part is a complete investigation. The authors could consider splitting the paper into two. Alternatively, the authors could focus on the first part, provide a full comparison to various clinical data, and show part 2 as a demonstration of model utility (and acknowledging the uncertainty regarding the real community situations).

Ultimately, we have chosen the latter route for the edits of this manuscript. Adding a significant portion to the fentanyl results has given a much broader picture to the reader of the strengths and weaknesses of the model. Future work will target the "what ifs" presented here such as variable dose timing, changing the recovery criteria, and adding variability to the CNS response of the patient.

Reviewer #2 (Remarks to the Author):

Interesting model of OIRD and naloxone reversal. Some items that need serious contemplation and adaptation:

1. The fentanyl model seems not appropriate, please address, particularly since the PK mismatch is appreciable - Fig 5-. Why not use a much simpler PK naloxone model with one or more absorption compartments??

We've included a much more detailed analysis of the fentanyl PK characteristics in the results section and have updated the model language to properly justify the use of the three compartment model detailed in the manuscript, line 289 highlighted with additional citations.

2. You mention "markers" in the abstract, but do not mention them again. What is meant by markers and why use this statement once?

Abstract is updated with details on the "markers" mentioned.

3. LM391 Fit to 2 mg dose: how and which parameter values had to be adjusted? First present the fit and later the validation results with other doses. Does Figure 4 for dose 2 mg represent the fit? The gray and purple drawings should be explained in the legend.

Great question, the fit was an adjustment of the kinetic parameters of the nasal deposition model. The text has been update to include this. Legend included in the naloxone pk curves for clarity.

4. LM 408 A deviation threshold of 20% is mentioned, and at LM 433 a 50% threshold - these seem arbitrary.

We've re-written and updated the paragraph describing the error between simulated and experimental data.

5. Figure 5: It would seem the mismatch between the fentanyl model and available data is quite large. What are the consequences for the conclusions of the paper?

beginning in line 432 we provide context for the fentanyl pk curve differences in the context of the given study and hope that this helps the reader understand some of the issues with the computational model when compared to experimental data.

6. Figures 8-12 Although the 2-min naloxone start is marked, is it correct that no naloxone is given? sorry if this wasn't clear, I've updated the language on line 516 to be more clear and figure 9-13

7. Figure 9: Why is there a sudden decrease in cardiac output after a period of almost linear increase? What do the spikes in the curves mean?

Beginning at line 625 we've added an updated description for figure 10, to help describe the dynamics occurring.

8. Figure 12: cm H2O as unit?

We report the unit in centimeter water but included language in the text to perform the conversion to mmhg if the reader would like.

9. LM 792: What does statistical significance mean here?

This has been clarified to explicitly call out the p-values for the computational experiments being performed for this paper (highlighted).

10. Fig. 13 Is there an indication of variability. Your suggestion of efficacy of 2 mg naloxone does not fit reality, it is much less effective than suggested by fig 13.

For this simulation, we only tested on a single patient, so variability wasn't present in the results. Investigations concerning populations of individuals (in a simulated setting) will be an area of future research. Patient variability in regards to respiratory depression and nervous system firing rates would be ideal stochastic parameters to introduce for that study but are not considered here. We note that 2mg naloxone dose is a standard rescue dose administer by first responders, as is a 4mg nasal spray. For these lower fentanyl values, this dose should be sufficient to reverse their symptoms, although other physiological consequences, which aren't considered in this study, such as reduced compliance of the lungs may provide complications to the resuscitation that aren't considered here with these small doses.

The main question that arises is whether the results of the simulated modeling is realistic and may be applied to humans?

We hope that these changes have given a clearer picture as to how this study should be interpreted. Ultimately it is a presentation of a whole-body physiological model of naloxone and fentanyl with some major limitations, discussed throughout the manuscript. It is only a first step towards creating a platform that could perform robust, validated, simulations of these patients (especially over populations). This manuscript is the first step in building a body of validity for the models. There is still a long way to go and we hope that we are honest in our assessment in the model performance and the real world impact of what is reported here. We sincerely thank the reviewer for their contributions, they have made the research much more robust.